# The Transcription Factors AcuK and AcuM Influence Siderophore Biosynthesis of *Aspergillus fumigatus*

**DOI:** 10.3390/jof10050327

**Published:** 2024-04-30

**Authors:** Patricia Caballero, Annie Yap, Michael J. Bromley, Hubertus Haas

**Affiliations:** 1Institute of Molecular Biology, Medical University of Innsbruck, Innrain 80-82, 6020 Innsbruck, Austria; patricia.caballero@i-med.ac.at (P.C.); yapannie2006@gmail.com (A.Y.); 2Manchester Fungal Infection Group, Division of Infection, Immunity, and Respiratory Medicine, The University of Manchester, Manchester M13 9PL, UK; mike.bromley@manchester.ac.uk

**Keywords:** fungi, *Aspergillus*, iron, uptake, siderophore, acetyl-CoA, gluconeogenesis, transcription factor

## Abstract

The mold *Aspergillus fumigatus* employs two high-affinity uptake systems, reductive iron assimilation (RIA) and siderophore-mediated iron acquisition (SIA), for the acquisition of the essential trace element iron. SIA has previously been shown to be crucial for virulence in mammalian hosts. Here, we show that a lack of AcuK or AcuM, transcription factors required for the activation of gluconeogenesis, decreases the production of both extra- and intracellular siderophores in *A. fumigatus*. The lack of AcuM or AcuK did not affect the expression of genes involved in RIA and SIA, suggesting that these regulators do not directly regulate iron homeostasis genes, but indirectly affect siderophore production through their influence on metabolism. Consistent with this, acetate supplementation reversed the intracellular siderophore production defect of Δ*acuM* and Δ*acuK*. Moreover, Δ*acuM* and Δ*acuK* displayed a similar growth defect under iron limitation and iron sufficiency, which suggests they have a general role in carbon metabolism apart from gluconeogenesis. In agreement with a potential role of the glyoxylate cycle in adaptation to iron starvation, transcript levels of the malate synthase-encoding *acuE* were found to be upregulated by iron limitation that is partially dependent on AcuK and AcuM. Together, these data demonstrate the influence of iron availability on carbon metabolism.

## 1. Introduction

The saprobic fungus *Aspergillus fumigatus* is the most common causative agent in a series of infectious diseases known as aspergillosis, which particularly impact immunocompromised patients [1,2]. Iron homeostasis has been shown to be key to the virulence of *A. fumigatus* in murine infection models [3]. *Aspergillus fumigatus* employs two high affinity iron uptake systems, reductive iron assimilation (RIA) and siderophore-mediated iron acquisition (SIA) [3]. RIA starts with the extracellular reduction in ferric iron to ferrous iron by the plasma membrane-localized metalloreductases such as FreB [4], followed by the re-oxidation and uptake of ferric iron employing the iron oxidase FetC and the ferric iron permease FtrA [5]. Siderophores are low molecular mass, ferric iron-specific chelators [6]. *A. fumigatus* produces two fusarinine-type siderophores termed triacetylfusarinine C (TAFC) and fusarinine C and two ferrichrome-type siderophores, termed ferricrocin (FC) and hydroxyferricrocin [3]. TAFC, fusarinine C, and ferricrocin are secreted to capture environmental iron; FC is also employed for intracellular iron handling within hyphae, and hydroxyferricrocin is employed for conidial iron storage [7,8,9,10]. The first committed step in siderophore biosynthesis is the formation of *N*^5^-hydroxyornithine from ornithine catalyzed by the monooxygenase SidA [5]. Subsequently, the pathways for the biosynthesis of fusarinine- and ferrichrome-type siderophores split due to the incorporation of different acyl-groups. The biosynthesis of fusarinine-type siderophores involves the mevalonyl-coenzyme A (CoA) ligase SidI, the mevalonyl-CoA hydratase SidH, the transacylase SidF, the nonribosomal peptide synthetases SidD and the transacetylase SidG [11,12]. The biosynthesis of ferrichrome-type siderophores involves the transacetylase SidL and the nonribosomal peptide synthetases SidC [12,13]. After the chelation of extracellular siderophores, ferric siderophores are taken up by specific siderophore-iron transporters: Sit1 and Sit2 for ferrichrome-type siderophores, MirD for fusarinine C, and MirB for TAFC [14,15,16]. The intracellular utilization of iron chelated by TAFC and fusarinine C involves siderophore hydrolysis mediated by SidJ and EstB [17,18,19].

The maintenance of iron homeostasis in *A. fumigatus* involves two iron-sensing transcription factors, termed as SreA and HapX, which act in cooperation with the CCAAT-binding complex [20,21,22]. During iron sufficiency, SreA represses SIA and RIA to avoid the toxicity caused by excessive iron uptake [23]. During iron starvation, HapX represses iron-consuming pathways such as the tricarboxylic acid (TCA) cycle, respiration, and heme biosynthesis as well as vacuolar iron deposition to spare iron and activates SIA and RIA [23]. Iron excess converts HapX into an activator of iron-dependent pathways and vacuolar iron deposition-mediated iron detoxification [3,24,25,26,27]. In invertebrate and murine aspergillosis models, the lack of siderophore biosynthesis (SidA) was shown to cause avirulence, while a lack of either fusarinine- (SidI, SidH, SidF, SidD) or ferrichrome-type (SidC) siderophores, a lack of TAFC uptake (MirB), and a lack of HapX-mediated iron regulation have been shown to attenuate virulence [3,28,29,30]. Moreover, the regulation of SIA and RIA involves *A. fumigatus* as well as transcription factors that do not directly sense the cellular iron state, such as LeuB, AtrR, and SrbA, which link iron regulation with leucine and sterol metabolism, and consequently to hypoxia adaptation [3,31,32]. Furthermore, the Zn_2_Cys_6_ transcription factors AcuM and AcuK, which are essential for the activation of gluconeogenesis in both *A. nidulans* and *A. fumigatus*, have been implicated in the regulation of RIA and SIA in *A. fumigatus* strain Af293 but not in *A. nidulans* [33,34,35]. However, AcuM and AcuK, which are both crucial for the full virulence of *A. fumigatus*, most likely function as a heterodimeric complex, they also appear to function independently of each other [33,34,35]. Essential gluconeogenic genes regulated by AcuM and AcuK include phosphoenolpyruvate carboxykinase-encoding *acuF*, fructose-1,6-bisphosphatase-encoding *acuG*, and enolase-encoding *acuN* [34,36]. In addition, AcuM and AcuK appear to regulate other carbon metabolic genes including TCA cycle enzymes.

To further characterize the underlying rational, we here aimed to analyze the role of AcuM and AcuK in the iron homeostasis of *A. fumigatus* strain A1160+. In agreement with previous reports [33,36], we found that the lack of AcuM or AcuK decreases the siderophore production. However, our data indicated that AcuM or AcuK impact siderophore biosynthesis indirectly rather than by the direct transcriptional control of genes involved in SIA. Possible explanations for the discrepancies are discussed.

## 2. Materials and Methods

### 2.1. Growth Conditions

For spore production, *A. fumigatus* strains were grown in *Aspergillus* complex media containing 20 mg/L Glucose (Glc) (Carl Roth, Karlsruhe, Germany), 1 g/L yeast extract (Lab M Limited, Rochdale, UK), 2 g/L peptone (Carl Roth GmbH, Karlsruhe, Germany), 1 g/L casamino acids (Sigma-Aldrich Inc., St. Louis, MO, USA), salt solution, and iron-free trace elements [37], at 37 °C for 5 days. For all other experiments, *Aspergillus* minimal medium containing either 1% (*w*/*v*) glucose or 1% (*w*/*v*) fructose as carbon source and either 20 mM glutamine (Gln) or 20 mM ammonium chloride (NH_4_^+^) as nitrogen source [37] was used; the trace element solution was prepared without iron salts and iron supplementation was performed as described. FeSO_4_ was added as indicated. Acetate supplemented cultures contained 5 mM acetate. For plate growth essays, 10^4^ spores of each strain was point-inoculated onto plates, which were incubated for 48 h at 37 °C. For liquid shake flask cultures, 100 mL were inoculated with 10^6^ spores per mL and incubated with 200 rpm at 37 °C for 18 h. 

### 2.2. A. fumigatus Strains Used

The mutant strains lacking AcuM or AcuK, Δ*acuM* and Δ*acuK*, were from the validated transcription factor deletion library [38] generated in *A. fumigatus* strain A1160+ (termed wt here), a derivative from the clinical strain *A. fumigatus* CEA10 lacking non-homologous recombination (ΔakuB^ku80^::*pyrG^−^zeo*, pyrG^−^::pyrG^Af^; MAT1-1) to facilitate genetic manipulation [39,40,41]. The *acuE* gene (AFUA_6g03540) was deleted by the replacement with the hygromycin resistance cassette (*hph*). Therefore, *acuE* flanking 5′- and 3′-non-coding regions were amplified from wt gDNA using the primers PC48/PC49 and PC50/51, respectively. The *hph* resistance cassette was amplified from plasmid pAN7-1 [42] using the primers hph_RC_fwd/hph_rev. The fragments were then assembled using fusion PCR with the primers PC52/PC53. The transformation of A. fumigatus was performed as described previously [43,44]. Transformants were isolated using minimal medium plates containing 0.1 mg/mL hygromycin B, resulting in the Δ*acuE* mutant, which was confirmed by Southern blot analysis (Appendix A). All strains used in this study are listed in Appendix A; primers and plasmids used for generation of the strains are listed in Appendix A. 

### 2.3. RNA Isolation and Northern Blot Analysis

RNA was isolated from the mycelium using TRI Reagent (Sigma-Aldrich Inc., St. Louis, MO, USA) as per manufacturer’s protocol. Ten micrograms of total RNA were separated using a 1.2% agarose gel with 1.85% (*w*/*v*) formaldehyde. The gel was blotted onto a Hybond^TM^-N+ membrane (Amersham Biosciences, Slough, UK). The detection by hybridization was performed using digoxigenin-labelled probes amplified by PCR. Primers used for the amplification of the Northern blot hybridization probes are listed in Appendix A.

### 2.4. Siderophore Analysis

Extraction and measurement of intracellular ferricrocin and extracellular TAFC was performed as previously described [21].

### 2.5. Statistical Analysis

Descriptive statistical analysis was performed with GraphPad Prism version 8.3.0 for Windows, GraphPad Software, San Diego, CA, USA, www.graphpad.com (31 January 2020).

## 3. Results

### 3.1. Lack of Either AcuM or AcuK Causes a Growth Defect under Both Iron Limitation and Sufficiency

To investigate a potential role of AcuM and AcuK in iron homeostasis, the growth of respective gene deletion mutants (Δ*acuM* and Δ*acuK*) was analyzed in comparison to the genetic background strain A1160+, a derivative of the clinical isolate CEA10 [40] and herein termed wt, with plate cultures reflecting different iron availability (Figure 1). In agreement with AcuM and AcuK being essential for gluconeogenesis [33,35,36], Δ*acuM* and Δ*acuK* mutant strains lacked growth when acetate and ammonium (NH_4_^+^) were used as carbon and nitrogen sources. In contrast, when glutamine (Gln) was used as the nitrogen source, the growth of the mutant strains was partially rescued as Gln can be used as an alternative, although not preferred, carbon source (Appendix A). 

With glucose (Glc) as a carbon source and Gln or NH_4_^+^ as a nitrogen source, both Δ*acuK* and Δ*acuM* mutant strains showed decreased growth under both iron starvation (−Fe), iron starvation in the presence of the ferrous iron-specific chelator bathophenathroline disulfonate (BPS), and iron sufficiency (+Fe) (Figure 1). BPS inhibits RIA and thereby emphasizes defects in SIA [5]. Appendix A shows that a siderophore-lacking Δ*sidA* mutant displays a growth defect under iron starvation in both the absence and the presence of BPS, which underlines the suitability of the used conditions to detect defects in the adaptation to iron starvation. Similarly, both Δ*acuM* and Δ*acuK* mutant strains displayed decreased biomass formation in liquid shake flask cultures under both iron starvation (−Fe) and iron sufficiency (+Fe) using either Gln or NH_4_^+^ as a nitrogen source (Figure 2A). Notably, when Gln was used as a nitrogen source, biomass formation was more severely impacted under iron sufficiency than iron depletion. The same was not observed when NH_4_^+^ was used as a nitrogen source, where decreases in biomass in strains lacking AcuM and AcuK were consistent under iron limitation and sufficiency. The biomass production of the wt strain during iron limitation was about 25% of that produced under iron sufficiency (Figure 2B), which clearly indicates iron starvation-mediated growth reduction. Notably, both Δ*acuM* and Δ*acuK* showed an about 10% increased −Fe/+Fe biomass ratio compared to wt with Gln as nitrogen source and a similar ratio with NH_4_^+^ as nitrogen source (Figure 2B). These data are in agreement with a general growth defect of the mutants rather than a defect in adaptation to iron starvation. 

### 3.2. Lack of Either AcuM or AcuK Decreases the Production of Both Extracellular and Intracellular Siderophores

As a next step, the production of extracellular and intracellular siderophores was analyzed; i.e., the TAFC content of the supernatants and the ferricrocin content of the aforementioned harvested biomass from the liquid shake flask cultures was compared (Figure 2C). Lack of AcuM decreased relative levels of TAFC to 46% and 48%, when accounting for biomass differences, as well as that of ferricrocin to 72% and 73%, with Gln and NH_4_^+^ as nitrogen source, respectively. The lack of AcuK decreased the production of TAFC to 64% and 48% as well as that of ferricrocin to 70% and 70% in these two nitrogen sources. Taken together, these data demonstrate that AcuM and AcuK are involved in siderophore production. 

### 3.3. Transcriptional Iron Regulation of RIA and SIA Is Largely Unaffected by Lack of AcuK or AcuM

Our data revealed an influence of AcuM and AcuK on siderophore production. Therefore, the potential implications of AcuM and AcuK in the transcriptional control of iron homeostasis was investigated by comparing the transcript levels of genes involved in iron acquisition in wt and mutants lacking AcuM or AcuK (Figure 3). Northern blot analyses indicated that the transcript levels of genes involved in RIA (iron permease-encoding *ftrA*), siderophore uptake (siderophore transporter-encoding *mirB*, *mirD*, and *sit1*) and siderophore biosynthesis (ornithine hydroxylase-encoding *sidA* and acyltransferase-encoding *sidF*) are largely unaffected by the lack of AcuM or AcuK: all investigated genes were similarly induced during iron limitation and downregulated during iron sufficiency in wt, Δ*acuM*, and Δ*acuK* strains. These data indicate that AcuK and AcuM are not involved in the transcriptional regulation of iron uptake systems in *A. fumigatus* A1160+, but rather indirectly impact siderophore production via metabolic effects.

### 3.4. Acetate Supplementation Cures the Defect in Ferricrocin Production Caused by Lack of AcuM or AcuK

Acetyl-CoA is a substrate for siderophore biosynthesis in *A. fumigatus*. It is directly used in ferricrocin biosynthesis and indirectly for the synthesis of anhydromevalonyl-CoA, which is required for the production of fusarinine-type siderophores such as TAFC [3]. Due to the inability of the Δ*acuM* and Δ*acuK* mutants to utilize acetate as a carbon source and their potential general role in carbon metabolism indicated by the growth defect with Glc as carbon source described above, we hypothesized that AcuM and AcuK might be involved in acetyl-CoA supply for growth and/or siderophore biosynthesis. Therefore, we analyzed the effect of acetate supplementation to a final concentration of 5 mM on growth and siderophore production using either Glc or fructose (Fru) as the main carbon source. Fru was used as a main carbon source to decrease the possible carbon catabolite repression effects mediated by Glc [45]. The Δ*acuM* and Δ*acuK* mutants were able to grow with Fru as the sole carbon source on solid media, displaying slightly decreased growth compared to wt similarly to using Glc as a carbon source (Figure 4). Moreover, Δ*acuM* and Δ*acuK* mutants showed defects in biomass formation, TAFC production and ferricrocin production with Fru as the sole carbon source similar to Glc as the sole carbon source in liquid shake flask cultures with NH4^+^ as the nitrogen source (Figure 5). The Δ*acuM* and Δ*acuK* mutants showed wt-like −Fe/+Fe biomass ratios with Fru similar to Glc as carbon source with and without acetate supplementation (Figure 5B). However, acetate supplementation cured the defect in the production of ferricrocin, but not TAFC, of Δ*acuM* and Δ*acuK* mutants irrespective of carbon source. Remarkably, acetate supplementation increased ferricrocin production by about 2.5-fold and 2-fold with Glc and Fru as the main carbon source, even in the wt (Figure 5). These data are in agreement with AcuK and AcuM, influencing at least ferricrocin production via metabolic control.

### 3.5. Transcriptional Regulation of Carbon Metabolism Is Affected by Iron Starvation but Is Only Partly Reliant on AcuM and AcuK

In a next step, the impact of AcuM and AcuK on the transcriptional regulation of selected carbon metabolic genes was investigated under iron limitation and sufficiency with Glc as a carbon source, and either Gln or NH_4_^+^ as a nitrogen source (Figure 6). The metabolic links of the enzymes encoded by these genes are shown in Appendix A. A scheme for the incorporation of acetyl-CoA and acetyl-CoA-derived anhydromevalonyl-CoA in the biosynthesis of TAFC and ferricrocin has been published previously [3].

The carboxykinase-encoding *acuF* and fructose-1,6-bisphosphatase-encoding *acuG* are essential gluconeogenic genes that have previously been shown to be directly regulated by AcuM and AcuK [34,36]. In agreement, the transcription of *acuF* was found to be strictly dependent on AcuM and AcuK during both iron sufficiency and iron limitation, as shown in Figure 6. The transcription of *acuG* was also strictly dependent on AcuM and AcuK during iron sufficiency; however, it appeared less reliant on both transcription factors during iron limitation. In contrast, the expression of gluconeogenetic enolase-encoding *acuN* was dependent on AcuM and AcuK mainly during iron limitation, but not sufficiency. The expression of isocitrate lyase-encoding *acuD*, which is essential for the glyoxylate cycle [46,47], was largely unaffected by iron availability in wt, but strictly dependent on AcuM and AcuK. The expression of another glyoxylate cycle gene, malate synthase-encoding *acuE*, showed upregulation during iron starvation in wt, and was partly dependent on AcuM and AcuK. Acetyl-CoA synthetase-encoding *facA*, which mediates the cytosolic acetyl-CoA production [48], and displayed upregulation during iron starvation independent of AcuM and AcuK, which is in agreement with increased acetyl-CoA requirement for siderophore production. The TCA cycle-involved aconitase-encoding *acoA* was downregulated during iron starvation and the expression was independent of AcuM and AcuK; this regulation was previously demonstrated to be mediated by the iron-sensing regulator HapX [25]. Taken together, these results demonstrate that iron limitation modulates the transcriptional regulation of carbon metabolism dependently and independently of AcuM and AcuK. Notably, the transcript levels of *acuM* and *acuK* were previously found to be similar during iron limitation, and insufficiency in a transcriptomic study [49], and *acuM* transcript levels were found to be similar during iron limitation and sufficiency even in the presence of an iron chelator [36].

### 3.6. Lack of AcuE Does Not Affect Growth Nor Siderophore Production

The iron limitation-induced upregulation of *acuE* in wt combined with a decreased expression in Δ*acuM* and Δ*acuK* (Figure 6) indicated a potential role of malate synthase in adaptation to iron starvation including siderophore biosynthesis. As expected [46,50], the inactivation of malate synthase by the deletion of the encoding gene (mutant strain Δ*acuE*) blocked the utilization of acetate as a carbon source (Figure 7). However, Δ*acuE* displayed wt-like growth on solid media with both Glc and Fru as carbon sources using either Gln or NH_4_^+^ as a nitrogen source under iron-limitation and sufficiency. When tested in a liquid shake flask culture, Δ*acuE* showed wt-like biomass formation during both iron limitation and sufficiency as well as the wt-like production of TAFC and ferricrocin with either Glc or Fru as the carbon source and NH_4_^+^ as the nitrogen source (Figure 8). 

## 4. Discussion

Here, we show that the lack of AcuM or AcuK in *A. fumigatus* strain A1160+ decreases the production of both extracellular TAFC and intracellular ferricrocin. Notably, the maximum decrease, dependent on the nitrogen source and the siderophore-type, did not exceed 48%, which indicates a modulation of siderophore production by these transcription factors. The transcript levels of genes involved in RIA, siderophore uptake, and siderophore biosynthesis, were largely unaffected by the lack of AcuM or AcuK, indicating that AcuK and AcuM are not involved in the transcriptional regulation of iron uptake systems in *A. fumigatus* A1160+, but rather indirectly impact the siderophore production via metabolic control. In agreement, acetate supplementation cured the intracellular siderophore production of Δ*acuM* and Δ*acuK*. However, acetate supplementation did not cure the TAFC production defect of Δ*acuM* and Δ*acuK*. In this respect, it is important to note that ferricrocin biosynthesis is based on the cytosolic use of acetyl-CoA for SidL-mediated acetylation of hydroxyornithine [13], while TAFC biosynthesis utilizes anhydromevalonyl-CoA, which is derived from acetyl-CoA via mevalonate synthesis, for the SidF-mediated acylation of hydroxyornithine within peroxisomes [51]. Consequently, the results indicate that AcuM and AcuK are involved in a cytosolic acetyl-CoA supply for ferricrocin production, which can be cured by acetate supplementation in contrast to TAFC biosynthesis taking place in peroxisomes. The Δ*acuM* and Δ*acuK* mutant strains displayed with Glc as a carbon source a similar growth defect under iron limitation and sufficiency, which suggests a general role in carbon metabolism apart from gluconeogenesis, rather than a direct role in direct iron regulation. In agreement, the chromatin-immunoprecipitation analysis showed the inter-dependent DNA binding of AcuM and AcuK to promoters of gluconeogenic genes in vivo independently of the carbon source in *A. nidulans* [8]. The impact of the metabolic flow of substrates from primary metabolism on secondary metabolites is a widely observed phenomenon in fungal metabolism [52,53]).

AcuM and AcuK are assumed to function mainly as part of the same complex [7,8]. Accordingly, phenotypes, siderophore production, and transcriptomic pattern were found to be highly similar between Δ*acuM* and Δ*acuK*. However, a study conducted in *A. fumigatus* Af293 indicated that AcuM and AcuK have partially independent functions of each other [7]. 

Previous reports using the *A. fumigatus* strain Af293 suggested the direct transcriptional regulation of iron homeostatic genes by AcuK or AcuM [33,36], which contrasts with the gene expression data presented herein. The differences might be explained by strain-specific effects. In this respect, it is interesting to note that Af293 shows a growth defect in liquid minimal medium shake flask cultures [54]. The studies with Af293 have been conducted with a complex medium, Sabouraud dextrose, employing the cell permeable iron chelator 1,10-phenanthroline, which is consequently able to chelate metals intracellularly [34,36]. In contrast, the studies described herein used to investigate siderophore production and the transcriptional control of iron homeostatic genes were performed without iron chelators to mimic natural iron limitation. Moreover, the chelator used here for some plate assays, BPS, is cell-impermeable to limit extracellular iron availability. Notably, both 1,10-phenanthroline and BPS are capable of chelating other metals, despite their preference for iron [55,56]. Consequently, chelators might affect metabolism in an iron-independent manner. Interestingly, AcuM has been reported to play no role in siderophore-mediated iron acquisition in *A. nidulans* [34,36]. In comparison, the transcriptional control of iron homeostasis by SreA and HapX is phylogenetically highly conserved [3]. 

The analysis of the transcript levels of selected carbon metabolic genes revealed that iron limitation impacts the regulation of carbon metabolism dependently and independently of AcuM and AcuK. As expected from previous studies identifying the targets of AcuM and AcuK [34,36], the transcription of gluconeogenetic carboxykinase-encoding *acuF* was found to be strictly dependent on AcuM and AcuK during iron sufficiency and iron limitation. However, the transcription of another previously identified target gene, gluconeogenetic fructose-1,6-bisphosphatase-encoding *acuG*, was dependent on AcuM and AcuK only during iron sufficiency, but was upregulated during iron limitation partly independently of AcuM and AcuK. In contrast, the expression of gluconeogenetic enolase-encoding *acuN* was dependent on AcuM and AcuK mainly during iron limitation but not sufficiency. The glyoxylate cycle essential gene *acuD*, which encodes isocitrate lyase [46,47], was largely unaffected by iron availability but strictly dependent on AcuM and AcuK. Another glyoxylate cycle gene, malate synthase-encoding *acuE*, showed upregulation during iron starvation in wt, which is partly dependent on AcuM and AcuK. Acetyl-CoA synthetase-encoding *facA*, which mediates cytosolic acetyl-CoA production [48], displayed upregulation during iron starvation independently of AcuM and AcuK, which is in agreement with the increased acetyl-CoA requirement for siderophore production. The TCA cycle-involved aconitase-encoding *acoA* showed downregulation during iron starvation independently of AcuM and AcuK; the iron regulation of *acoA* was previously demonstrated to be mediated by the iron-sensing regulator HapX [25]. The iron limitation-induced upregulation of *acuE*, partially dependently on AcuK and AcuM, indicated a possible role of the encoding enzyme in iron homeostasis and/or siderophore biosynthesis. As expected from previous studies [46,50], the inactivation of AcuE blocked the utilization of acetate as a carbon source but did not impact either growth under iron limitation and sufficiency nor siderophore production. These data suggest that AcuE is either not involved in iron starvation adaptation or that its absence is compensated by other metabolic pathways, such as fumarase-mediated malate synthesis in the TCA cycle. Nevertheless, these results demonstrated the influence of iron as well as AcuM and AcuK on carbon metabolism with Glc as carbon source.

## Figures and Tables

**Figure 1 jof-10-00327-f001:**
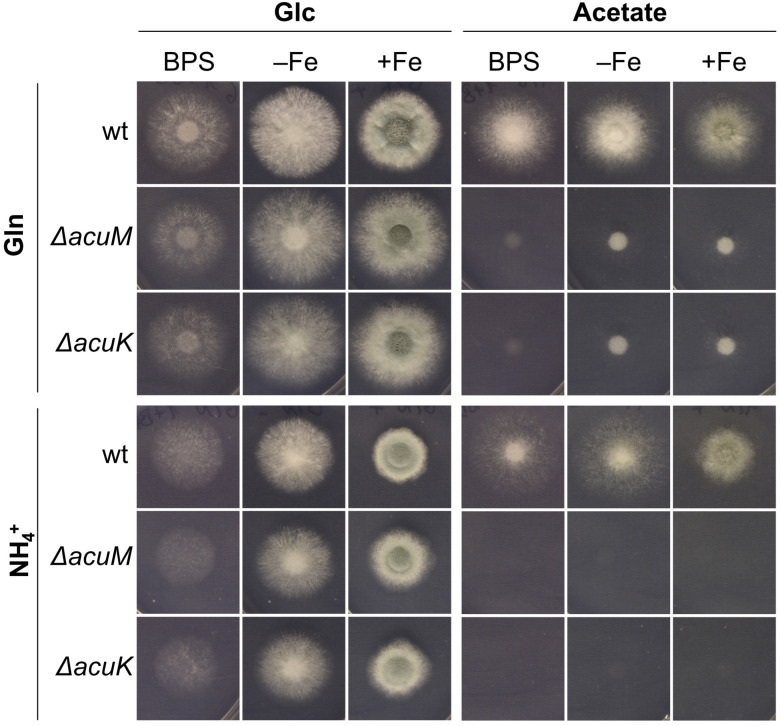
Lack of either AcuM or AcuK blocks growth on acetate and decreases growth with Glc as the carbon source on solid media. *Aspergillus fumigatus* wt and Δ*acuM* and Δ*acuK* mutant strains were point-inoculated using 10^4^ spores on minimal medium using either Glc or acetate as carbon sources and Gln or NH_4_^+^ as nitrogen sources. Media reflected iron limitation (−Fe), iron limitation in the presence of the iron-specific chelator BPS (1 µM FeSO_4_ plus 0.2 mM BPS) (BPS), and iron sufficiency (30 µM FeSO_4_). Plates were incubated at 37 °C for 48 h.

**Figure 2 jof-10-00327-f002:**
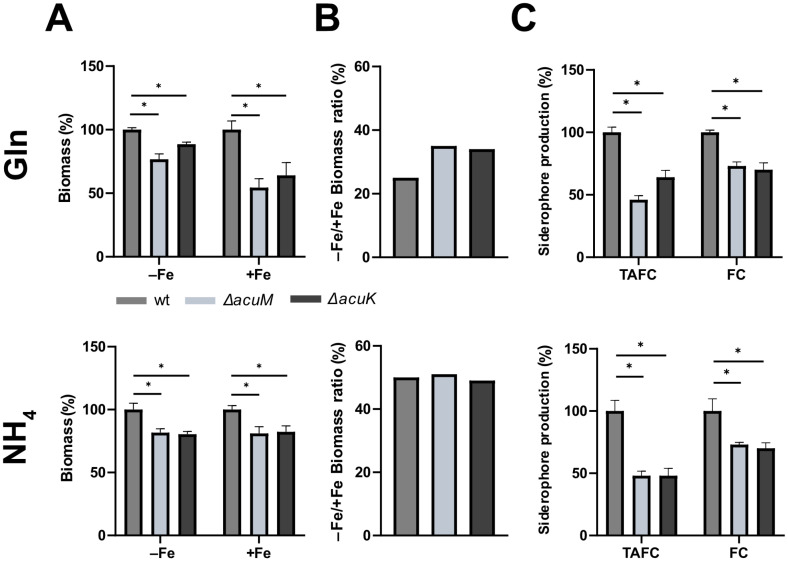
Lack of either AcuM or AcuK decreases liquid growth under both iron starvation and sufficiency, and reduces siderophore production. The 100 mL shake flask cultures were inoculated with 1 × 10^8^ conidia of respective fungal strains. The used minimal medium contained Glc as a carbon source and either Gln or NH_4_^+^ as the nitrogen source. The media were either supplemented with 30 µM FeSO_4_ (+Fe) or lacked the addition of iron (−Fe). After incubation at 37 °C for 18 h, the biomass and supernatant of each culture was collected; biomass was measured after freeze-drying and normalized to that of wt grown under the same conditions (**A**). The −Fe/+Fe ratio of the biomass of each strain is shown in (**B**). TAFC was extracted from the collected supernatants and ferricrocin (FC) from the freeze-dried biomasses from iron limited cultures; values were normalized to that of wt grown under the same conditions. (**C**) Siderophore production under iron limitation normalized to biomass and wt. Siderophore production was not detected under iron sufficiency. The mean values ± SD of biological triplicates are shown and statistically significant differences by *t*-Student test are indicated by * (*p* ≤ 0.005). Absolute values are given in Appendix A.

**Figure 3 jof-10-00327-f003:**
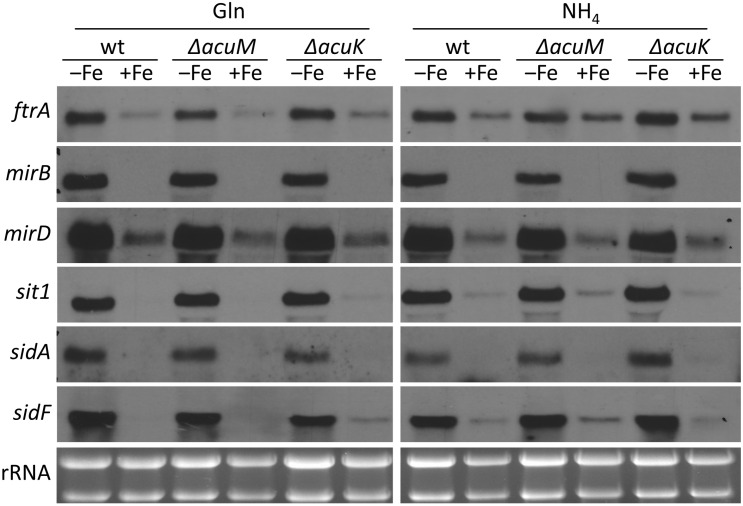
Transcriptional regulation of RIA and SIA is largely independent of AcuM and AcuK. Shake flask cultures were performed as described in Figure 2. Northern blot analysis was performed with isolated total RNA. rRNA served as a control for quality and loading of the RNA samples.

**Figure 4 jof-10-00327-f004:**
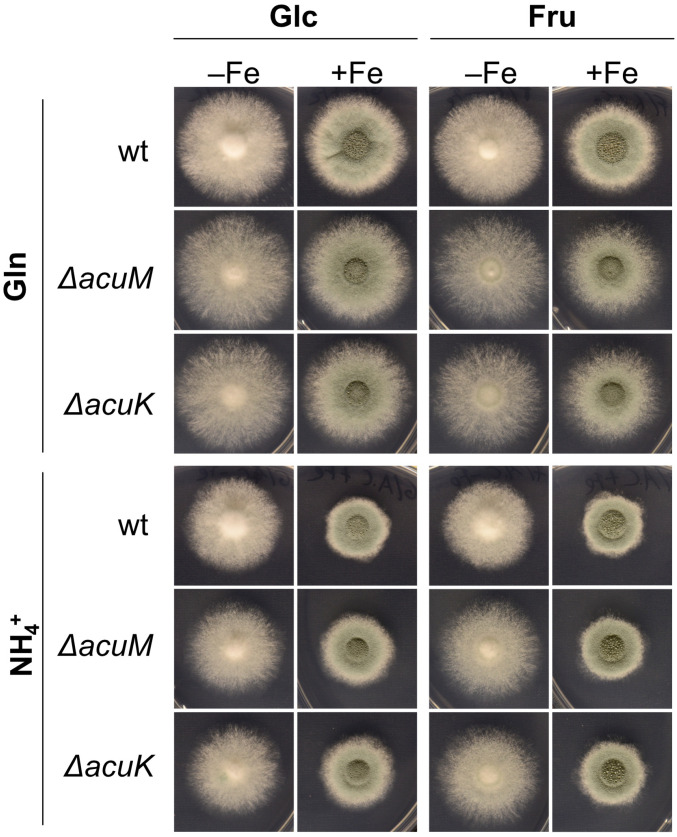
Lack of AcuM or AcuK causes a similar growth defect with Glc and Fru as the sole carbon source. Experimental details are described in Figure 1.

**Figure 5 jof-10-00327-f005:**
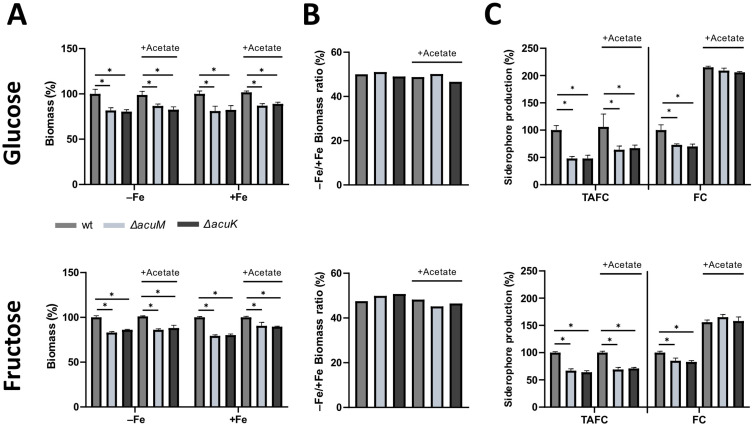
Acetate supplementation cures the ferricrocin production defect caused by lack of AcuM or AcuK but does not affect the defects in growth or TAFC production. Cultures contained NH_4_^+^ as the sole nitrogen source and either Glc or Fru as the sole carbon source. Cultures supplemented with 5 mM acetate are labeled with “+Acetate”. Experimental details are described in Figure 2. For a better comparison, the Glc culture results without acetate supplementation from Figure 2 are included. The biomass and siderophore values of acetate-supplemented cultures were normalized to the wt grown under the same conditions without acetate supplementation. Absolute values are given in Appendix A. (**A**) Biomass (%) measured after freeze-drying and normalized to that of wt grown under the same conditions; (**B**) −Fe/+Fe ratio of the biomass of each strain; (**C**) Siderophore production under iron limitation normalized to biomass and wt. The mean values ± SD of biological triplicates are shown and statistically significant differences by *t*-Student test are indicated by * (*p* ≤ 0.005).

**Figure 6 jof-10-00327-f006:**
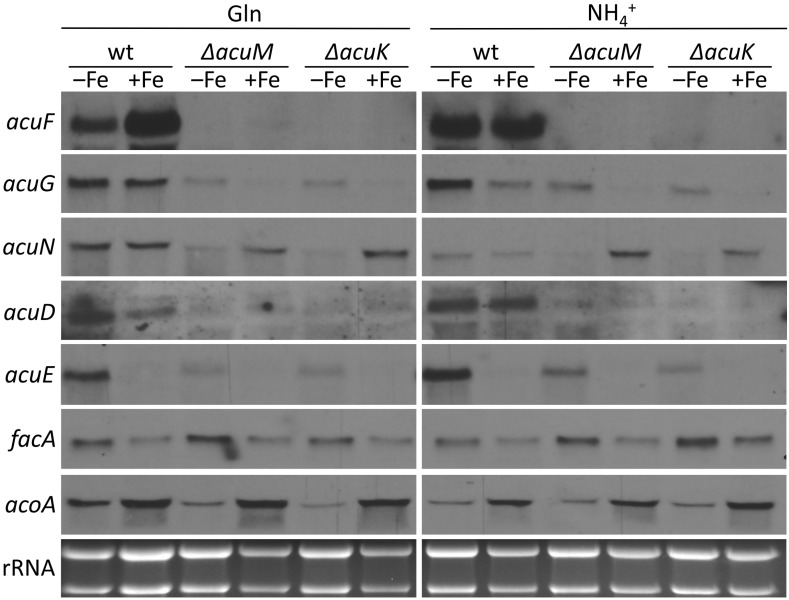
Transcriptional regulation of carbon metabolism is affected by iron starvation dependently and independently of AcuM and AcuK. Shake flask cultures were performed as described in Figure 2 with Glc as the carbon source and either Gln or NH_4_^+^ as the nitrogen source. Northern blot analysis was performed with isolated total RNA. rRNA served as a control for quality and loading of the RNA samples.

**Figure 7 jof-10-00327-f007:**
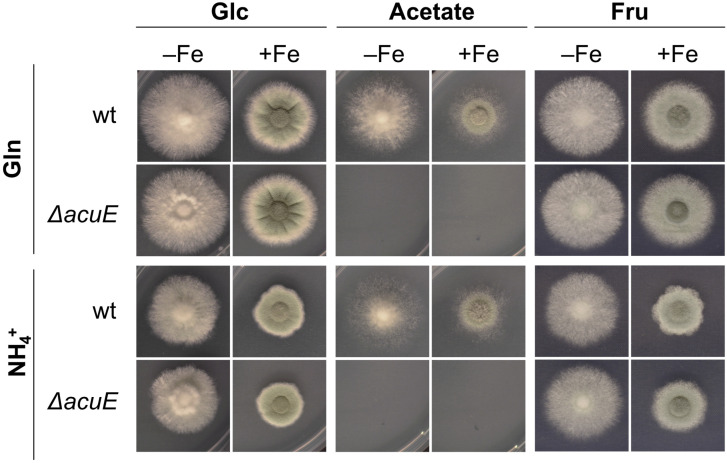
Lack of AcuE blocks the utilization of acetate as a carbon source. Experimental details are described in Figure 1 using either Glc or acetate as the carbon source and either Gln or NH_4_^+^ as the nitrogen source.

**Figure 8 jof-10-00327-f008:**
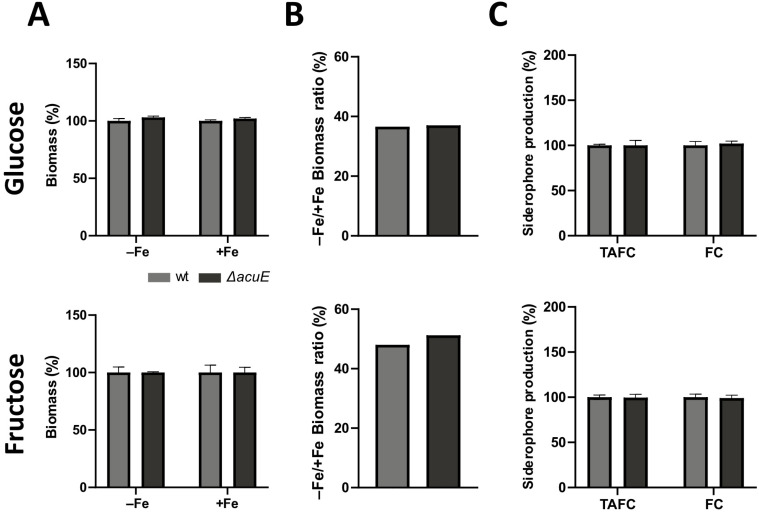
Lack of AcuE affects neither growth under iron limitation and sufficiency nor production of TAFC or ferricrocin. Cultures contained either Glc or Fru as the sole carbon source and NH_4_^+^ as the sole nitrogen source. Experimental details are described in Figure 2. Absolute values are given in Appendix A. (**A**) Biomass (%) measured after freeze-drying and normalized to that of wt grown under the same conditions (**B**) −Fe/+Fe ratio of the biomass of each strain; (**C**) Siderophore production under iron limitation normalized to biomass and wt.

## Data Availability

Data are contained within the article and Appendix A.

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
