# Peer review of "The Transcription Factors AcuK and AcuM Influence Siderophore Biosynthesis of Aspergillus fumigatus"

_jof, 2024, doi:10.3390/jof10050327_

Round 1
Reviewer 1 Report
The work of Haas´s lab investigates the impact of the gene deletions of the AcuM and AcuK transcription factors in the human pathogen Aspergillus fumigatus on a) growth behavior and b) siderophore production. Both biomass formation and siderophore production (TAFC and FC) were slightly reduced. The authors show that expression of RIA (reductive iron uptake) or SIA (siderophore mediated iron uptake) genes are not affected by the gene deletions. The authors suggested the diminished production of siderophores might be result of a reduced metabolic flow of acetate/acetyl-CoA in the mutants, which was indeed the case for at least FC. Data is clearly presented and was thoroughly analyzed. However, the current version provides only marginal novel information to the fungal research community as a similar work was prepared almost 10 years ago: The impact on growth and siderophore production of the A. fumigatus acuM and acuK deletion mutants (and a double deletion mutant) in presence of various carbon sources and under iron sufficient and limited conditions have been shown by Fillers and Shepards group in 2015:
Pongpom, Monsicha et al. “Divergent targets of Aspergillus fumigatus AcuK and AcuM transcription factors during growth in vitro versus invasive disease.” Infection and immunity vol. 83,3 (2015): 923-33. doi:10.1128/IAI.02685-14.
Hence, the impact of AcuM and AcuK on siderophore production is known for Aspergillus fumigatus. Moreover, the impact of acuM and acuK on gluconeogenesis/ glycoxylate cycle has been demonstrated for various ascomycetes as well (https://doi.org/10.3390/jof7100798, https://doi.org/10.1111/j.1365-2958.2012.08067.x, https://doi.org/10.1099/00221287-92-2-263).
Hence, at its current state of the manuscript, additional experiments are required to create novelty, e.g. the acetyl-CoA metabolism in various compartments of the cell (as discussed in l. 301-309) might be investigated.
l. 77-82: The authors introduce acuD here, but they do not describe the function of acuD.
l. 222: How do the authors explain the rescued production of FC, but not TAFC by acetate supplementation? Compartments? Are there acetyl-CoA transporters involved or regulated?
l.239-263: To ease the reader to follow the authors thread of thoughts, I recommend to add an additional scheme here showing the metabolic regulation and metabolic/transcriptional network of the investigated proteins that were figured out during this expression analysis.
l. 303: anydrovevalonyl-CoA must read anhydromevalonyl-CoA
Figure 1: Iron limitation is sometimes a difficult task as iron is a ubiquitous contaminant in nutritional compounds (such as glucose or yeast extract). May the authors show – in addition to their acuM and acuK mutant – a verified iron susceptible mutant (delta-hapX or delta-sidA) that clearly shows a growth defect on iron limitation?
Northern Blots:
Most Northern Blots are of best quality. However, the Blot for acuD contains artefacts and might be repeated to provide a picture of higher quality.
Supporting information:
Tables S1-S3: I really appreciate that the authors share the raw data here. May the units for biomass (grams?) and siderophore compounds (µg?) additionally provided?
Table S6: Why didn´t the authors test for acuM and acuK additionally? Are they deregulated under iron-limited and sufficient conditions?
Reviewer 2 Report
In the manuscript entitled "The Transcription Factors AcuK and AcuM Influence Siderophore Biosynthesis of Aspergillus fumigatus", the authors characterized the function of Zn2Cys6 transcription factors AcuM and AcuK in the human pathogenic fungus A. fumigatus. Through exploring phenotypic, molecular and biochemical experiments the role of these proteins in siderophore biosynthesis is explored. The scientific design is appropriate and conclusions drawn from this are excellent. I do have several comments that should be addressed before this is fit for publication:
Major:
- The assay method of siderophore is required in the “Materials and Methods” section or any other suitable section.
- The Fig. 6 is very unclean. Please changed to more clear one.
Minor:
- Please check all fungus name as Italic.
Reviewer 3 Report
The manuscript by Caballero et al. reports the influence of transcriptional factors AcuK and AcuM on siderophore biosynthesis. The data presented here suggest that the influence is indirect, presumably via carbon metabolism. The work is rather straightforward and simple, and conducted in an old-fashioned way, for example, northern blotting rather than transcriptomics analsis used more commonly now. The work has merits, but does not substantially advance current understanding. Nevertheless, the results are clearly presented and this reviewer has one major comment.
Acetate supplementation reverses the intracellular siderophore production defect of ΔacuM and ΔacuK, but the inability to utilize acetate as observed in the acuE mutant does not affect siderophore production. Authors suggest that AcuE is either not involved in iron starvation adaptation or that its absence is compensated by other metabolic pathways. At least in the latter case, further discussion should be done. What could other metabolic pathways be?
The manuscript is well-written in general. But there are some confusing sentences and typos.
For example, line 204, "Acetyl-CoA is an essential cofactor for siderophore biosynthesis in A. fumigatus as it is directly used in ferricrocin biosynthesis and indirectly, as a precursor of anhydromevelonyl-CoA, in biosynthesis of fusarinine-type siderophores such as TAFC biosynthesis". Acetyl-CoA is a substrate rather than a cofactor.
Line 291, the lack of
"Δ“s in mutants, such as ΔacuM, are written in both italic and regular in the text and legends (Fig. 6). should be written consistently throughout the text.
Round 2
Reviewer 1 Report
I thank the authors for the revised version of the manuscript.
Comment 1:
We do not agree with the lack of novelty of our study as we do not present the same results as reported previously. AcuM and AcuK have been previously published to regulate siderophore production by direct regulation of transcript levels of the respective genes in A. fumigatus strain Af293. Moreover, the lack of these transcription factors was shown to cause a growth defect under iron limitation. Our studies did not confirm these results for A. fumigatus strain A1160+; neither the growth defect during iron limitation caused by lack of these transcription factors nor their direct influence on transcript levels of genes involved in high-affinity iron acquisition. The only overlap is the decreased siderophore production caused by AcuM/K inactivation. The difference between the two studies might indicate strain-specific effects, which we discussed (lines 318-333 in the original manuscript, now 327-340). We also discussed differences in the growth pattern of these two A. fumigatus strains (strain Af293 shows in contrast to A1160+ a growth defect in liquid minimal medium culturing: Ref 51 and Refs therein) and the differences in the experimental set up: in the previous publications a cell-permeable chelator was used, which might lead to artificial results, while the key experiments in the current study were done without chelators (lines 318-333 in the original manuscript, now 327-340). We avoid chelators as they are not absolutely metal-specific and might therefore cause indirect effects. We used the same experimental set up, which we used for characterization of numerous iron-related cellular components (reviewed in Ref 3). We think that it is important to report that the results from the previous studies cannot be generalized for all A. fumigatus strains, at least not for A1160+.
Furthermore, we show that acetate supplementation cures the defect in ferricrocin production, which is in line with a metabolic effect of AcuM/K inactivation on siderophore production. In addition, we demonstrate an impact of iron limitation of expression of carbon catabolic genes (Fig. 6) and show that inactivation of AcuE, a carbon metabolic gene found to be transcriptionally upregulated by iron limitation in an AcuM/K dependent manner, does influence neither growth under iron limitation nor siderophore production.
Response:
I thank the authors for pointing this out. I agree with the authors that work on different strains may cause different out-comes and should be kept in mind during research on pathogenicity mechanisms. This is especially of interest in terms of rising antifungal resistance of several clinical Aspergillus strains, which should be briefly highlighted here. Please cite the respective literature.
Comment 2:
We are sorry for the mistake:” AcuD” should read ”AcuK” here! Thank you for pointing out!
Corrected!
Response: Thank you.
Comment 3:
Indeed, we believe that the difference are due to involvement of different compartments: we discussed this in the original version (lines 301-307, now lines 312-318):
”In this respect important to note: ferricrocin biosynthesis is based on cytosolic use of acetyl-CoA for
SidL-mediated acetylation of hydroxyornithine [13], while TAFC biosynthesis utilizes anydrovevalonyl-CoA, which is derived from acetyl-CoA via mevalonate synthesis, for SidF-mediated acylation of hydroxyornithine within peroxisomes [50]. Consequently, the results indicate that AcuM and AcuK are involved in cytosolic acetyl-CoA supply for ferricrocin production, which can be cured by acetate supplementation in contrast to TAFC biosynthesis.” We now added ”taking place in peroxisomes” at the end of the last sentence (line 314).
Response:
Thank you. I recommend to include references here showing that metabolic flow of substrates from primary metabolism impacts the quantity of secondary metabolites which is a widely observed phenomenon in microbial metabolism, e.g in
antimicrobial peptide biosynthesis of Penicillium rubens (e.g. 10.1093/jimb/kuab045),
diterpenoid biosynthesis of the blast fungus Magnaporthe oryzae (e.g. doi.org/10.1111/j.1365-313X.2010.04264.x ), or
microcystin production in cyanobacteria (e.g. doi.org/10.1016/j.aquatox.2023.106525 ).
Comment 4:
We added a scheme displaying the metabolic interaction of the enzymes encoded by the analyzed genes as Supplementary Fig. S4, which is cited in the text (lines 249-250).
Response:
Thank you for the additional figure. However, I suggest to incorporate the connection of primary (gluconeogenesis) with secondary metabolism (siderophore production) as well, maybe starting from the displayed acetyl-CoA.
Comment 5:
Thank you for pointing out – corrected here and at other spots.
Response: Thank you.
Comment 6:
As suggested, we added a growth assay comparing wt and a mutant lacking sidA showing that the used conditions identify mutants with defects in adaptation to iron limitation: Supplementary Fig. S3 cited in the text (lines 149-152).
Response:
Thank you for the additional figure.
Comment 7:
We agree that the acuD Northern blot is not of the best quality. However, it shows similar results with both Gln and NH4+ as nitrogen source and therefore we think that it is trustworthy. The first author of the study is currently unavailable for several weeks not available. In order to meet the proposed timeline for submission of the revision, have not repeated this analysis. We are of course willing to do so if the Reviewer deems it essential.
Response:
I agree. Although not of highest quality, the RNA bands for acuD are clearly visible and the Blot does not need to be repeated. It was just a recommendation.
Comment 8:
We have added the information as suggested.
Response: Thank you.
Comment 9:
Transcript levels of acuM and acuK were previously found to be similar during iron limitation and sufficiency in a transcriptome study (PMID: 18721228). Furthermore, transcript levels of acuM were found to be similar during iron limitation and sufficiency even in the presence of an iron chelator (Ref 36).
Response: Thank you for the additional information. Please indicate the constitutive expression of acuM and acuK in the manuscript referring to the suggested literature.
Line 225: NH4+ : Please use subscripts/superscripts here.
